# Measurement-induced, spatially-extended entanglement in a hot, strongly-interacting atomic system

Jia Kong[1,2 ✉], Ricardo Jiménez-Martínez[2], Charikleia Troullinou[2], Vito Giovanni Lucivero [2], Géza Tóth [3,4,5,6] & Morgan W. Mitchell[2,7 ✉]

Quantum technologies use entanglement to outperform classical technologies, and often employ strong cooling and isolation to protect entangled entities from decoherence by random interactions. Here we show that the opposite strategy—promoting random interactions—can help generate and preserve entanglement. We use optical quantum non-demolition measurement to produce entanglement in a hot alkali vapor, in a regime dominated by random spin-exchange collisions. We use Bayesian statistics and spin-squeezing inequalities to show that at least $1.52(4) \times 10^{13}$ of the $5.32(12) \times 10^{13}$ participating atoms enter into singlet-type entangled states, which persist for tens of spin-thermalization times and span thousands of times the nearest-neighbor distance. The results show that high temperatures and strong random interactions need not destroy many-body quantum coherence, that collective measurement can produce very complex entangled states, and that the hot, strongly-interacting media now in use for extreme atomic sensing are well suited for sensing beyond the standard quantum limit.

[1] Department of Physics, Hangzhou Dianzi University, 310018 Hangzhou, China. [2] ICFO–Institut de Ciencies Fotoniques, The Barcelona Institute of Science and Technology, 08860 Castelldefels (Barcelona), Spain. [3] Department of Theoretical Physics, University of the Basque Country UPV/EHU, P.O. Box 644, E-48080 Bilbao, Spain. [4] Donostia International Physics Center, E-20018 San Sebastián, Spain. [5] IKERBASQUE, Basque Foundation for Science, E-48011 Bilbao, Spain. [6] Wigner Research Centre for Physics, Hungarian Academy of Sciences, P.O. Box 49, H-1525 Budapest, Hungary. [7] ICREA–Institució Catalana de Recerca i Estudis Avançats, 08010 Barcelona, Spain. ✉email: jia.kong@hdu.edu.cn; morgan.mitchell@icfo.eu

Entanglement is an essential resource in quantum computation, simulation, and sensing[1], and is also believed to underlie important many-body phenomena such as high-$T_c$ superconductivity[2]. In many quantum technology implementations, strong cooling and precise controls are required to prevent entropy—whether from the environment or from noise in classical parameters—from destroying quantum coherence. Quantum sensing[3] is often pursued using low-entropy methods, for example, with cold atoms in optical lattices[4]. There are, nonetheless, important sensing technologies that operate in a high-entropy environment, and indeed that employ thermalization to boost coherence and thus sensor performance. Notably, vapor-phase spin-exchange-relaxation-free (SERF) techniques[5] are used for magnetometry[6,7], rotation sensing[8], and searches for physics beyond the standard model[9], and give unprecedented sensitivity[10]. In the SERF regime, strong, frequent, and randomly-timed spin-exchange (SE) collisions dominate the spin dynamics, to produce local spin thermalization. In doing so, these same processes also decouple the spin degrees of freedom from the bath of centre-of-mass degrees of freedom, which increases the spin coherence time[5]. Whether entanglement can be generated, survive, and be observed in such a high entropy environment is a challenging open question[11].

Here, we study the nature of spin entanglement in this hot, strongly-interacting atomic medium, using techniques of direct relevance to extreme sensing. We apply optical quantum non-demolition (QND) measurement[12,13]—a proven technique for both generation and detection of non-classical states in atomic media—to a SERF-regime vapor. We start with a thermalized spin state to guarantee the zero mean of the total spin variable and use a [1, 1, 1] direction magnetic field (see Fig. 1a) to achieve QND measurements on three components of the total spin variable. We track the evolution of the net spin using the Bayesian method of Kalman filtering[14], and use spin squeezing inequalities[15,16] to quantify entanglement from the observed statistics. We observe that the QND measurement generates a macroscopic singlet state[17]—a squeezed state containing a

macroscopic number of singlet-type entanglement bonds. This shows that QND methods can generate entanglement in hot atomic systems even when the atomic spin dynamics include strong local interactions. The spin squeezing and thus the entanglement persist far longer than the spin-thermalization time of the vapor; any given entanglement bond is passed many times from atom to atom before decohering. We also observe a sensitivity to gradient fields that indicates the typical entanglement bond length is thousands of times the nearest-neighbor distance. This is experimental evidence of long-range singlet-type entanglement bonds. These experimental observations complement recent predictions of coherent inter-species quantum state transfer by spin collision physics[18,19].

## Results

**Material system.** We work with a vapor of $^{87}$Rb contained in a glass cell with buffer gas to slow diffusion, and housed in magnetic shielding and field coils to control the magnetic environment, see Fig. 1a. The density is maintained at $n_{Rb} = 3.6 \times 10^{14}$ atoms/cm³, and the magnetic field, applied along the [1, 1, 1] direction, is used to control the Larmor precession frequency $\omega_L/2\pi$. At this density, the spin-exchange collision rate is $325 \times 10^3$ s⁻¹. For $\omega_L$ below about $2\pi \times 5$ kHz, the vapor enters the SERF regime, characterized by a large increase in spin coherence time.

**Spin thermalization.** The spin dynamics of such dense alkali vapors[5] is characterized by a competition of several local spin interactions, diffusion, and interaction with external fields, buffer gases, and wall surfaces. While the full complexity of this scenario has not yet been incorporated in a quantum statistical model, in the SERF regime an important simplification allows us to describe the state dynamics in sufficient detail for entanglement detection, as we now show.

If $\mathbf{j}^{(l)}$ and $\mathbf{i}^{(l)}$ are the $l$th atom's electron and nuclear spins, respectively, the spin dynamics, including sudden collisions, can

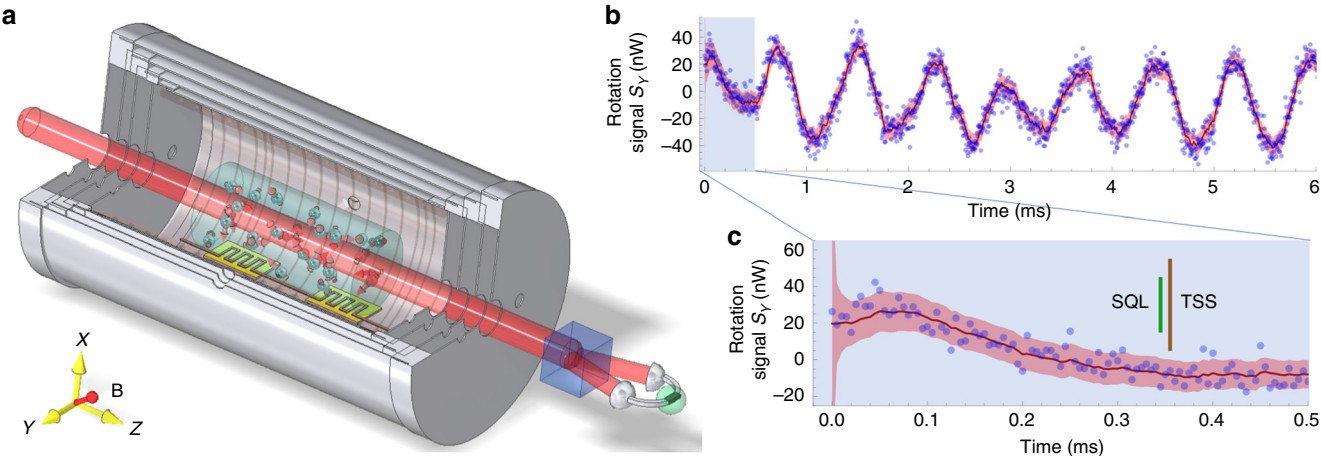

**Fig. 1 Experimental principle. a** Experimental setup. A linearly polarized probe beam, red detuned by 44 GHz from the $^{87}$Rb D₁ line, passes through a glass cell containing a hot $^{87}$Rb vapor and 100 torr of N₂ buffer gas, which is housed in a low-noise magnetic enclosure. The transmitted light is detected with a shot-noise-limited polarimeter (a Wollaston prism plus differential detector), which indicates $\mathcal{F}_z$, the projection of the collective spin $\mathcal{F}$ on the probe direction, plus optical shot noise. A static magnetic field along the [1, 1, 1] direction causes the spin components to precess as $\mathcal{F}_z \to \mathcal{F}_x \to \mathcal{F}_y$ every one-third of a Larmor cycle. In this way the polarimeter record contains information about all three components[17]. **b** Representative sample of the Stokes parameter $S_y^{(out)}(t)$ showing raw data (blue dots), optimal estimate for the atomic spin $g\mathcal{F}_z(t)S_x$ (red line), and $\pm 4\sigma$ confidence interval (pale red region), as computed by Kalman filter (KF). Signal clearly shows atomic spin coherence over ms time-scales. **c** Expanded view of early signal, colors as in (**b**), bars show $\pm 4\sigma$ confidence regions of $g\mathcal{F}_z(t)S_x$ for a thermal spin state (TSS) and the standard quantum limit (SQL) for a spin-polarized state. The KF acquires a sub-SQL estimate for $\mathcal{F}_z$ in 20 µs, far less than the coherence time. Probe power = 2 mW, Larmor frequency = 1.3 kHz, cell temperature = 463 K.

be described by the time-dependent Hamiltonian

$$H = \hbar A_{hf} \sum_l \mathbf{j}^{(l)} \cdot \mathbf{i}^{(l)} + \hbar \sum_{ll'n} \theta_n \delta\left(t - t_n^{(l,l')}\right) \mathbf{j}^{(l)} \cdot \mathbf{j}^{(l')}$$
$$+ \hbar \sum_{lm} \psi_m \delta\left(t - t_m^{(l)}\right) \mathbf{j}^{(l)} \cdot \mathbf{d}_m^{(l)} + \hbar \gamma_e \sum_l \mathbf{j}^{(l)} \cdot \mathbf{B} \quad (1)$$

where the terms describe the hyperfine interaction, SE collisions, spin-destruction (SD) collisions, and Zeeman interaction, respectively. $A_{hf}$ is the hyperfine (HF) splitting and $t_n^{(l,l')}$ is the (random) time of the $n$-th SE collision between atoms $l$ and $l'$, which causes mutual precession of $\mathbf{j}^{(l)}$ and $\mathbf{j}^{(l')}$ by the (random) angle $\theta_n$. We indicate with $R_{SE}$ the rate at which such collisions move angular momentum between atoms. Similarly, the third term describes rotations about the random direction $\mathbf{d}_m$ by random angle $\psi_m$, and causes spin depolarization at a rate $R_{SD}$. $\gamma_e = 2\pi \times 28$ GHz T$^{-1}$ is the electron spin gyromagnetic ratio. We neglect the much smaller $\mathbf{i} \cdot \mathbf{B}$ coupling. We note that short-range effects of the magnetic dipole–dipole interaction (MDDI) are already included in $R_{SE}$ and $R_{SD}$, and that long-range MDDI effects are negligible in an unpolarized ensemble, as considered here.

The SERF regime is defined by the hierarchy $A_{hf} \gg R_{SE} \gg \gamma_e|B|$, $R_{SD}$. Our experiment is in this regime, as we have $A_{hf} \approx 10^9$ s$^{-1}$, $R_{SE} \approx 10^5$ s$^{-1}$, $\gamma_e|B| \approx 10^4$ s$^{-1}$ and $R_{SD} \approx 10^2$ s$^{-1}$. The hierarchy implies the following dynamics: on short times, the combined action of the HF and SE terms rapidly thermalizes the spin state, i.e., generates the maximum entropy consistent with the ensemble total angular momentum $\mathbf{F}$, which is conserved by these interactions (see "Methods", "Spin thermalization" section). We indicate this $\mathbf{F}$-parametrized max-entropy state by $\rho_{\mathbf{F}}^{(th)}$. We note that entanglement can survive the thermalization process; for example, $\rho_{F=0}^{(th)}$ is a singlet and thus necessarily describes entangled atoms. On longer time-scales, $\mathbf{F}$ experiences precession about $\mathbf{B}$ due to the Zeeman term and diffusive relaxation due to the depolarization term.

**Non-destructive measurement.** We perform a continuous non-destructive readout of the spin polarization using Faraday rotation of off-resonance light. On passing through the cell the optical polarization experiences rotation by an angle $g\mathcal{F}_z(t) \ll \pi$, where $z$ is the propagation axis of the probe, $g$ is a light-atom coupling constant and $\mathcal{F} \equiv \mathbf{F}_a - \mathbf{F}_b$, where $\mathbf{F}_\alpha$ is the collective spin orientation from atoms in hyperfine state $\alpha \in \{1, 2\}$ (see "Methods", "Observed spin signal" section).

For thermalized spin states $\langle \mathcal{F} \rangle \propto \langle \mathbf{F} \rangle$, so that the observed polarization rotation gives a view into the full spin dynamics. The optical rotation is detected by a balanced polarimeter (BP), which gives a signal proportional to the Stokes parameter

$$S_y^{(out)}(t) = S_y^{(in)}(t) + g\mathcal{F}_z(t)S_x, \quad (2)$$

where $S_x$ is the Stokes component along which the input beam is polarized[20]. $S_y^{(in)}(t)$ is a zero-mean Gaussian process, whose variance is dictated by photon shot-noise and is characterized by a power-spectral analysis of the BP signal[21].

**Spin dynamics and spin tracking.** The evolution of $\mathcal{F}(t)$ is described by the Langevin equation (see "Methods", "Spin dynamics" section)

$$d\mathcal{F} = \gamma\mathcal{F} \times \mathbf{B}dt - \Gamma\mathcal{F}dt + \sqrt{2\Gamma Q}d\mathbf{W} \quad (3)$$

where $\gamma = \gamma_e/q$ is the SERF-regime gyromagnetic ratio, i.e., that of a bare electron reduced by the nuclear slowing-down factor[5], which takes the value $q = 6$ in the SERF regime[22]. $\Gamma$ is the net relaxation rate including diffusion, spin-destruction collisions, and probe-induced

decoherence, $Q$ is the equilibrium variance (see below) and $dW_h$, $h \in \{x, y, z\}$ are independent temporal Wiener increments.

Based on Eqs. (3) and (2), we employ the Bayesian estimation technique of Kalman filtering (KF)[14] to recover $\mathcal{F}(t)$, which is shown as $g\mathcal{F}_z(t)S_x$ to facilitate comparison against the measured $S_y^{(out)}(t)$ in Fig. 1b). The KF (see "Methods", "Kalman filter" section) gives both a best estimate and a covariance matrix $\Gamma_{\mathcal{F}}(t)$ for the components of $\mathcal{F}(t)$, which gives an upper bound on the variances of the post-measurement state. Figure 1c shows that the $\mathcal{F}_z$ component of $\Gamma_{\mathcal{F}}(t)$ is suppressed rapidly, to reach a steady state value which is below the SQL. The other components are similarly reduced in variance by the measurement, and the total variance $|\Delta\mathcal{F}|^2 \equiv \mathrm{Tr}[\Gamma_{\mathcal{F}}]$ can be compared against spin squeezing inequalities[15,16] to detect and quantify entanglement: Defining the spin-squeezing parameter $\xi^2 \equiv |\Delta\mathcal{F}|^2/\mathrm{SQL}$, where $\mathrm{SQL} \equiv N_A 13/8$ is the standard quantum limit, $\xi^2 < 1$ detects entanglement, indicating a macroscopic singlet state[17]. The minimum number of entangled atoms[15] is $N_A(1 - \xi^2)13/16$ (see "Methods", "Entanglement witness" section).

**Experimental results.** The cell temperature was stabilized at 463 K to give an alkali number density of $n_{Rb} = 3.55(6) \times 10^{14}$ atoms cm$^{-3}$, calibrated as described in "Methods", "Density calibration" section, and thus $N_A = 5.32(12) \times 10^{13}$ atoms within the $3$ cm $\times$ $0.0503(8)$ cm$^2$ effective volume of the beam. At this density, the SE collision rate is $R_{SE} \approx 325 \times 10^3$ s$^{-1}$. By varying $B$ we can observe the transition to the SERF regime, and the consequent development of squeezing. Figure 2a shows spin-noise spectra (SNS)[21], i.e., the power spectra of detected signal from BP, for different values of $B$, from which we determine the resonance frequency $\omega_L = \gamma B$, relaxation rate $\Gamma$ and the number density. Using these as parameters in the KF (see "Methods", "Kalman filter" section), we obtain $|\Delta\mathcal{F}|^2$ as shown in Fig. 2b, including a transition to squeezed/entangled states as the system enters the SERF regime.

At a Larmor frequency of 1.3 kHz, we observe $\xi^2 = 0.650(2)$ or $1.88(1)$ dB of spin squeezing at optimal probe power 2 mW (see "Methods", "Kalman filter" section), which implies that at least $1.52(4) \times 10^{13}$ of the $5.32(12) \times 10^{13}$ participating atoms have become entangled as a result of the measurement. This greatly exceeds the previous entanglement records: $5 \times 10^5$ cold atoms in singlet states using a similar QND strategy[17] and a Dicke state involving $2 \times 10^{11}$ impurities in a solid, made by storing a single photon in a multi-component atomic ensemble[23]. This is also the largest number of atoms yet involved in a squeezed state; see Bao et al. for a recent record for polarized spin-squeezed states[24]. We use this power and field condition for the experiments described below, and note that the spin-relaxation time greatly exceeds the spin-thermalization time. In this condition, the entanglement bonds are rapidly distributed amongst the atoms by SE collisions without being lost.

We now study the spatial distribution of the induced entanglement. As concerns the observable $\mathcal{F}$, the relevant dynamical processes, including precession, decoherence, and probing, are permutationally-invariant: Eqs. (3) and (2) are unchanged by any permutation of the atomic states. This suggests that any two atoms should be equally likely to become entangled, and entanglement bonds should be generated for atoms separated by $\Delta z \in [0, L]$, where $L = 3$ is the length of the cell. Indeed, such permutational invariance is central to proposals[25,26] that use QND measurement to interrogate and manipulate many-body systems. There are other possibilities, however, such as optical pumping into entangled sub-radiant states[27], that could produce localized singlets.

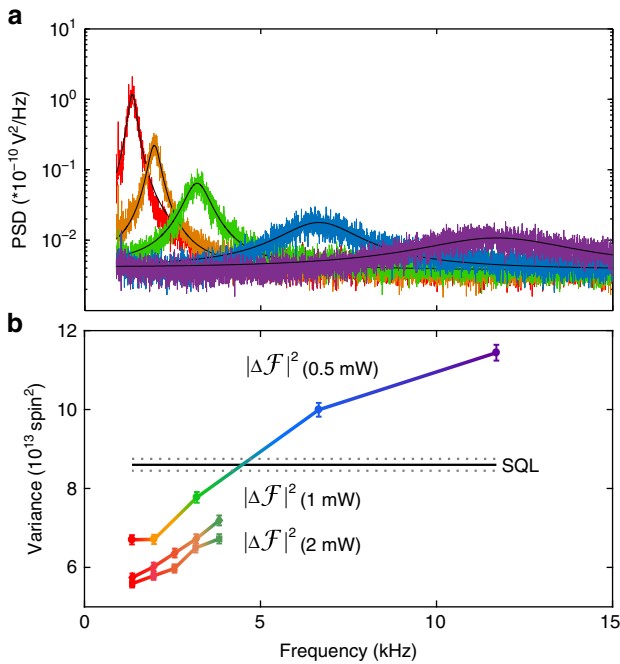

**Fig. 2 Quantum non-demolition detection of collective spin in the strongly-interacting regime. a** Spin noise spectra with atomic spin signal driven by thermal fluctuations and precessing at the Larmor frequency ($\nu_L = \omega_L/2\pi$) rising above shot noise of the Faraday rotation probe. Different spectra correspond to different bias field strengths. Black lines are single Lorentz fits for the spectra. Comparing the red and purple curves, we see a roughly 100-fold improvement in signal to noise ratio (SNR) due to suppression of SE relaxation and consequent line narrowing, which indicated a stronger quantum non-demolition (QND) interaction. **b** Spin variance versus Larmor frequency. Black solid-line shows the standard quantum limit of total spin (SQL = $N_A 13/8$). Dashed horizontal lines indicate standard quantum limit (SQL) $\pm 1\sigma$ statistical uncertainty. Round symbols show $|\Delta\mathcal{F}|^2$ measured with 0.5 mW probe light, corresponding to the spectra in (**a**). Diamonds and squares show $|\Delta\mathcal{F}|^2$ measured with 1 mW and 2 mW probe light respectively. All error bars show $\pm 1\sigma$ uncertainty due to uncertainty in atomic number, including uncertainties in atomic density and effective volume (see "Methods", "Density calibration" section).

We test for long-range singlet-type entanglement by applying a weak gradient $B' \equiv d|B|/dz$ during the cw probing process. A magnetic field gradient, if present, causes differential Larmor precession that converts low-noise singlets into high-noise triplets, providing evidence of long-range entanglement. For example, singlets with separation $\Delta z$ will convert into triplets and back at angular frequency[28] $\Omega = \gamma B' \Delta z$. The range $\delta\Delta z$ of separations then induces a range $\delta\Omega = \gamma B' \delta\Delta z$ of conversion frequencies, which describes a relaxation rate. In Fig. 3 we show the KF-estimated $|\Delta\mathcal{F}_z|^2$ as a function of $B'$ and of time since the last data point, which clearly shows faster relaxation toward a thermal spin state with increasing $B'$. The observed additional relaxation for $B' = 57.2$ nT mm$^{-1}$ (relative to $B' = 0$) is $\delta\Omega = 1.54 \times 10^3$ s$^{-1}$, found by an exponential fit. For $\Delta z$ on the order of a wavelength, as would describe sub-radiant states, we would expect $\delta\Omega \sim 1$ s$^{-1}$ at this gradient, which clearly disagrees with observations. The observed r.m.s. separation $\delta\Delta z$ is about one millimeter, which is thousands of times the typical nearest-neighbor distance $n_{Rb}^{-1/3} \approx 0.14\,\mu$m.

## Discussion

Our observation of complex, long-lived, spatially-extended entanglement in SERF-regime vapors has a number of implications.

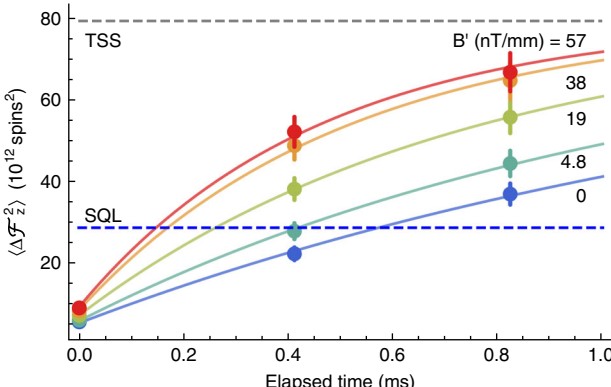

**Fig. 3 Evidence for long-range entanglement.** Points show the KF-obtained variances $\langle\Delta\mathcal{F}_z^2\rangle$ of the rotating-frame spin component $\mathcal{F}_z$ as a function of delay since $\mathcal{F}_z$ was last aligned along the laboratory z axis, and thus subject to measurement by Faraday rotation. Error bars show $\pm 4\sigma$ uncertainty, originating in the uncertainty of atom number. As seen in these data, increased gradient $B'$ causes a faster relaxation toward the thermal spin state (TSS) value, as is expected for a gas of singlets with a range of separations (along the z axis) in the mm range. The blue and gray dashed lines are the standard quantum limit (SQL) ($2.88(7) \times 10^{13}$ spins$^2$) and TSS ($7.99(18) \times 10^{13}$ spins$^2$) noise levels, respectively. Probe power = 2 mW.

First, it is a concrete and experimentally tractable example of a system in which entanglement is not only compatible with, but, in fact, stabilized by entropy-generating mechanisms—in this case strong, randomly-timed spin-exchange collisions. It is particularly intriguing that the observed macroscopic singlet state shares several traits with a spin liquid state[2], which is conjectured to underlie high-temperature superconductivity, a prime example of quantum coherence surviving in an entropic environment. Second, the results show that optical quantum non-demolition measurement can efficiently produce complex entangled states with long-range entanglement. This confirms a critical assumption of QND-based proposals[25,26] for QND-assisted quantum simulation of exotic antiferromagnetic phases. Third, the results show that SERF media are compatible with both spin squeezing and QND techniques, opening the way to quantum enhancement of what is currently the most sensitive approach to low-frequency magnetometry and other extreme sensing tasks.

## Methods

**Density calibration**. In the SERF regime, and in the low spin polarization limit, decoherence introduced by SE collisions between alkali atoms is quantified by[5,29,30]

$$\pi\Delta\nu_{SE} = \omega_L^2 \frac{2I[-3 + I(1 + 4I(I+2))]}{3[3 + 4I(I+1)]R_{SE}},\qquad(4)$$

where for $^{87}$Rb atomic samples the nuclear spin $I = 3/2$, and $\omega_L = \gamma_e|\mathbf{B}|/q$. In Eq. (4) the spin-exchange collision rate $R_{SE} = \sigma_{SE}n_{Rb}\overline{V}$ is proportional to the alkali density $n_{Rb}$ with proportionality dictated by the SE collision cross-section $\sigma_{SE}$ and the relative thermal velocity between two colliding $^{87}$Rb atoms $\overline{V}$. Using the reported value[31] of $\sigma_{SE} = 1.9 \times 10^{-14}$ cm$^2$ and $\overline{V} = 4.75 \times 10^4$ cm s$^{-1}$, which is computed for $^{87}$Rb atoms at a temperature of 463 K, we then calibrate the alkali density by fitting the measured linewidth $\Delta\nu$ as a function of $\omega_L$. The model uses $\Delta\nu = \Delta\nu_0 + \Delta\nu_{SE}$, where $\Delta\nu_{SE}$ is given by Eq. (4), and $\Delta\nu_0$ describes density-independent broadening due to power broadening and transit effects. $n_{Rb}$ and $\Delta\nu_0$ are free parameters found by fitting, with results shown in Fig. 4.

**Observed spin signal**. For a collection of atoms, we define the collective total atomic spin $\mathbf{F} \equiv \sum_l \mathbf{f}^{(l)}$, where $\mathbf{f}^{(l)}$ is the total spin of the $l$th atom. We identify the contributions of the two hyperfine ground states $F_a = 1$ and $F_b = 2$, defined as $\mathbf{F}_\alpha \equiv \sum_l \mathbf{f}_\alpha^{(l)}$, where $\mathbf{f}_\alpha^{(l)}$ describes the contribution of atoms in state $F_\alpha$, such that $\mathbf{f}^{(l)} = \mathbf{f}_a^{(l)} + \mathbf{f}_b^{(l)}$.

The Faraday rotation signal arises from an off-resonance coupling of the probe light to the collective atomic spin. To lowest order in $\mathbf{F}$, as appropriate to the regime of the experiment, the polarization signal $S_y$ is related to the collective spin

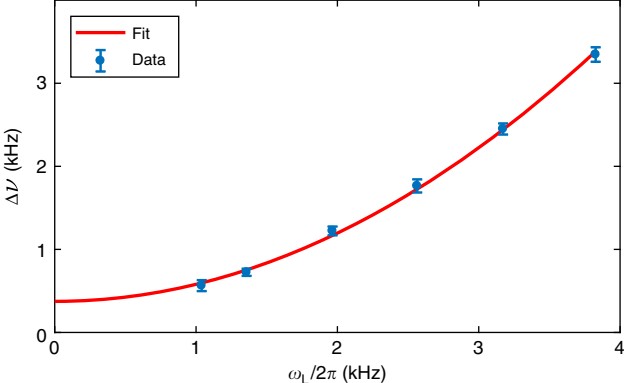

**Fig. 4 Density calibration by spin noise spectroscopy.** Graph shows the full-width at half maximum linewidth $\Delta\nu$ as a function of $\omega_L$, across the transition into the spin-exchange relaxation free (SERF) regime. Blue dots and error bars show the mean $\pm 1\sigma$ standard error of the mean, obtained from the fitting of 20 spectra. Red line shows Eq. (4) fit to the data with density $n_{Rb}$ as a free parameter. Probe power = 1 mW, T = 463 K.

variables $F_{a,z}$, $F_{b,z}$ through the input-output relation[20,32–34]

$$S_y^{(out)}(t) \approx S_y^{(in)}(t) + \left(g_a F_{a,z} - g_b F_{b,z}\right) S_x^{(in)}(t), \tag{5}$$

where $S_\alpha \equiv (E_+^{(-)}, E_-^{(-)})\sigma_\alpha(E_+^{(+)}, E_-^{(+)})^T/2$ are Stokes operators, $\sigma_\alpha$, $\alpha \in \{x, y, z\}$ are the Pauli matrices and $E_\beta^{(\pm)}$ is the positive-frequency (negative-frequency) part of the quantized electromagnetic field with polarization $\beta = \pm$ for sigma-plus (sigma-minus) polarized light. The factor $(g_a F_{a,z} - g_b F_{b,z}) \equiv \Theta_{FR}$ plays the role of a Faraday rotation angle, which in this small-angle regime can be seen to cause a displacement of $S_y(t)$ from its input value. It should be noted that $\Theta_{FR}$ is operator-valued, enabling entanglement of the spin and optical polarizations, and that the hyperfine ground states $F_a$, $F_b$ contribute differentially to it.

The coupling constants are[14,33,35]

$$g_\alpha = \frac{1}{2I+1} \frac{cr_e f_{osc}}{A_{eff}} \frac{\nu - \nu_\alpha}{(\nu - \nu_\alpha)^2 + (\Upsilon/2)^2}, \tag{6}$$

where $r_e = 2.82 \times 10^{-13}$ cm is the classical electron radius, $f_{osc} = 0.34$ is the oscillator strength of the $D_1$ transition in Rb, $c$ is the speed of light, and $\nu - \nu_\alpha$ is the optical detuning of the probe-light. $\Upsilon = 2.4$ GHz is the pressure-broadened full-width at half-maximum (FWHM) linewidth of the $D_1$ optical transition for our experimental conditions of 100 Torr of $N_2$ buffer gas. For a far-detuned probe beam, such that $|\nu - (\nu_a + \nu_b)/2| \gg |\nu_a - \nu_b|$ (as in this experiment) one can approximate $g \equiv g_a \approx g_b$, such that

$$\Theta_{FR} \approx g(\mathbf{F}_a - \mathbf{F}_b)_z \equiv g\boldsymbol{\mathcal{F}}_z. \tag{7}$$

**Spin thermalization**. In a local region containing a mean number of atoms $N_A$, the SE and HF mechanisms will rapidly produce a thermal state $\rho$. We note that this process conserves $\mathbf{F}$, and thus also conserves the statistical distribution of $\mathbf{F}$, including possible correlations with other regions. $\rho$ is then the maximum-entropy state consistent with a given distribution of $\mathbf{F}$. Partitioning arguments then show that, for weakly polarized states such as those used in this experiment, the mean hyperfine populations are $\langle N_a \rangle / N_A = 3/8$ and $\langle N_b \rangle / N_A = 5/8$ and the polarisations are $\langle \mathbf{F}_a \rangle = \langle \mathbf{F} \rangle / 6$, $\langle \mathbf{F} \rangle_b = \langle \mathbf{F} \rangle 5/6$, from which the FR signal is $\langle \Theta_{FR} \rangle = g\langle \boldsymbol{\mathcal{F}}_z \rangle = -g\langle F \rangle_z 2/3$. The same relations must hold for spin observables that sum $\mathbf{F}$ over larger regions, including the region of the beam, which determines which atoms contribute to the observed signal.

**Entanglement witness**. We can construct a witness for singlet-type entanglement[16] as follows: we define the total variance

$$|\Delta\boldsymbol{\mathcal{F}}|^2 \equiv \text{var}(\boldsymbol{\mathcal{F}}_x) + \text{var}(\boldsymbol{\mathcal{F}}_y) + \text{var}(\boldsymbol{\mathcal{F}}_z). \tag{8}$$

Separable states of $N_A$ atoms will obey a limit $|\Delta\boldsymbol{\mathcal{F}}|^2 \geq N_A\mathcal{C}$, where $\mathcal{C}$ is a constant, meaning that $|\Delta\boldsymbol{\mathcal{F}}|^2 < N_A\mathcal{C}$ witnesses entanglement. To find $\mathcal{C}$, we note that a product state of $N_a$ atoms in state $F_a$ and $N_b$ atoms in state $F_b$ has $|\Delta\boldsymbol{\mathcal{F}}|^2 \geq \sum_\alpha N_\alpha F_\alpha$. Separable states are mixtures of product states. For such states, due to the concavity of the variance, $|\Delta\boldsymbol{\mathcal{F}}|^2 \geq \sum_\alpha \langle N_\alpha \rangle F_\alpha$ holds[16]. In light of the 3:5 ratio resulting from spin thermalization, this gives

$$|\Delta\boldsymbol{\mathcal{F}}|^2 \geq \frac{3}{8}N_A + 2\frac{5}{8}N_A = \frac{13}{8}N_A, \tag{9}$$

or $\mathcal{C} = 13/8$. Therefore, the standard quantum limit (SQL) is $N_A 13/8$. We define the degree of squeezing $\xi^2 \equiv |\Delta\boldsymbol{\mathcal{F}}|^2 / (\sum_\alpha \langle N_\alpha \rangle F_\alpha)$. Meanwhile the "thermal spin state (TSS)," i.e. the fully-mixed state, has $|\Delta\boldsymbol{\mathcal{F}}|^2 = N_A 9/2$.

Our condition provides also a quantitative measure of the number of entangled atoms. We consider a pure entangled quantum state of the form

$$|\kappa\rangle = |\psi^{(1)}\rangle \otimes |\psi^{(1)}\rangle \otimes |\psi^{(2)}\rangle \otimes \dots \otimes |\psi^{(N_p)}\rangle \otimes |\Phi_e\rangle, \tag{10}$$

where $|\psi^{(l)}\rangle$ are single particle states. Here, $N_p$ particles are in a product state, while $N_e = N_A - N_p$ particles are in an entangled state denoted by $|\Phi_e\rangle$. For the collective variances of $|\kappa\rangle$ we can write that

$$(\Delta\boldsymbol{\mathcal{F}}_h)_\kappa^2 = \sum_{l=1}^{N_p} (\Delta\boldsymbol{\mathcal{F}}_h)_{\psi^{(l)}}^2 + (\Delta\boldsymbol{\mathcal{F}}_h)_{\Phi_e}^2 \tag{11}$$

for $h = x, y, z$. Let us try to find a lower bound on (11).

Let us assume that all atoms are in state $F_\alpha$. Then, we know that $(\Delta\boldsymbol{\mathcal{F}}_h)_{\psi^{(l)}}^2 \geq F_\alpha$ while $(\Delta\boldsymbol{\mathcal{F}}_h)_{\Phi_e}^2$ can even be zero, if the entangled state $|\Phi_e\rangle$ is a perfect singlet. Hence,

$$|\Delta\boldsymbol{\mathcal{F}}|^2 \geq N_p F_\alpha \equiv (N_A - N_e)F_\alpha. \tag{12}$$

Based on these, the number of entangled atoms in this case is bounded from below as $N_e \geq (1 - \xi^2)N_A$, where $\xi^2 = |\Delta\boldsymbol{\mathcal{F}}|^2 / \text{SQL}$ and the standard quantum limit SQL in this case is $F_\alpha N_A$.

Let us now consider the case when some atoms have $F_1$ others have $F_2$. In particular, let us consider a state of the type (10) such that $N_\alpha$ particles have spin $F_\alpha$ with $\alpha = 1, 2$ such that $F_1 \leq F_2$. Then, for such a pure state,

$$(\Delta\boldsymbol{\mathcal{F}}_x)^2 + (\Delta\boldsymbol{\mathcal{F}}_y)^2 + (\Delta\boldsymbol{\mathcal{F}}_z)^2 \geq nF_1 + (N_p - n)F_2 \tag{13}$$

holds, where

$$n = \min(N_p, N_1). \tag{14}$$

Note that the bound in Eq. (13) is sharp, since it can be saturated by a quantum state of the type (10). In order to minimize the left-hand side of Eq. (13), the particles corresponding to the product part must have as many spins in $F_1$ as possible, since this way we can obtain a small total variance. In particular, if $N_P \geq N_1$ then all atoms in the product part must have an $F_1$ spin, otherwise at least $N_1$ atoms of the $N_p$ atoms.

It is instructive to rewrite Eq. (10) with a piece-wise linear bound as

$$(\Delta\boldsymbol{\mathcal{F}}_x)^2 + (\Delta\boldsymbol{\mathcal{F}}_y)^2 + (\Delta\boldsymbol{\mathcal{F}}_z)^2 \geq \begin{cases} N_1 F_1 + (N_p - N_1)F_2, & \text{if } N_p > N_1 \\ N_p F_1, & \text{if } N_p \leq N_1 \end{cases} \tag{15}$$

The bound in Eq. (15) is plotted in Fig. 5a.

So far, we have been discussing a bound for a pure state of the form (10). The results can be extended to a mixture of such states straightforwardly, since the bound in Eq. (13) is convex in $(N_1, N_p)$. Then, in our formulas $N_1$, must be replaced by $\langle N_1 \rangle$. We also have to define the number of entangled particles $N_e$ for the case of a mixed state. A mixed state has $N_e$ entangled particles, if it cannot be constructed as a mixture of pure states, which all have fewer than $N_e$ entangled particles[15].

We know that in our experiments $F_1 = 1$, $F_2 = 2$, and $\langle N_1 \rangle = 3/8 N_A$. From these, we obtain the minimum number of entangled atoms as

$$N_e \geq \begin{cases} \frac{13}{16}(1 - \xi^2)N_A, & \xi^2 \geq \frac{3}{13}, \\ (1 - \frac{13}{8}\xi^2)N_A, & \xi^2 < \frac{3}{13}. \end{cases} \tag{16}$$

The bound in Eq. (16) is plotted in Fig. 5b. Here, again $\xi^2 = |\Delta\boldsymbol{\mathcal{F}}|^2 / \text{SQL}$ and the standard quantum limit SQL in this case is $N_A \times 13/8$. For our experiment, the $\xi^2 \geq 3/13$ case is relevant.

The macroscopic singlet state gives a metrological advantage in estimating gradient fields[28,36] and in detecting displacement of the spin state, e.g. by optical pumping[14,37].

**Balanced polarimeter signal**. The photocurrent $I(t)$ of the balanced polarimeter shown in Fig. 1a is

$$I(t) = \Re \int_A dx dy\, S_y^{(out)}(x, y, t), \tag{17}$$

where the detector's responsivity is $\Re = q_e\eta/E_{ph}$ in terms of the detector quantum efficiency $\eta$, charge of the electron $q_e$, and photon energy $E_{ph}$. To account for its spatial structure in Eq. (17) the integral is carried over the area of the probe. From Eq. (2) and Eq. (17) one obtains the differential photocurrent increment

$$I(t)dt = \eta g' \dot{N}\boldsymbol{\mathcal{F}}_z(t)dt + dw_{sn}(t), \tag{18}$$

where $g' = gq_e$, the stochastic increment $dw_{sn}(t)$, due to photon shot-noise, is given by $dw_{sn}(t) = \sqrt{\eta q_e^2 \dot{N}}dW$ with $\dot{N}$ being the photon-flux and $dW \sim \mathcal{N}(0, dt)$

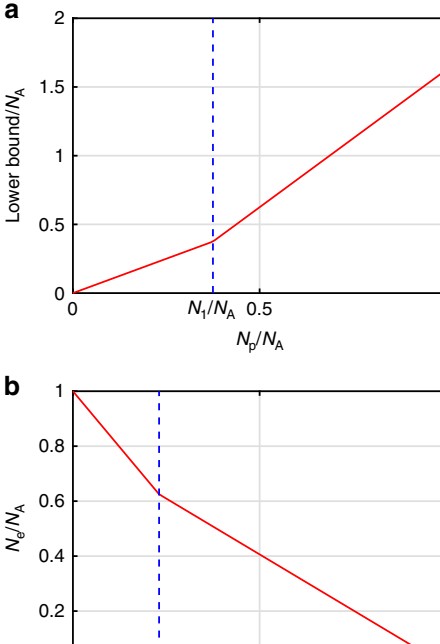

**Fig. 5 Entanglement witness. a** Lower bound on the sum of the three variances given in Eq. (13) as a function of $N_p$ for a quantum state of the type given in Eq. (10). We set $N_1/N_A = 3/8$, $F_1 = 1$, $F_2 = 2$, corresponding to the experiment. If we had only particles with the same spin, it would just be a straight line. **b** The lower bound on the number of entangled spins, given in Eq. (16), as a function of the spin-squeezing parameter $\xi^2$.

representing a differential Wiener increment. In our experiments the photocurrent $I(t)$ is sampled at a rate $\Delta^{-1} = 200\,\text{k}$ Samples/s. To formulate the discrete-time version of Eq. (17) we consider the sampling process as a short-term average of the continuous-time measurement. The photocurrent $I(t_k)$ recorded at $t_k = k\Delta$, with $k$ being an integer, can then be expressed as

$$I(t_k) = \frac{1}{\Delta} \int_{t_k-\Delta}^{t_k} I(t')dt' = \eta g' \dot{N} \mathcal{F}_z(t_k) + \xi_D(t_k), \qquad (19)$$

where the Langevin noise $\xi_D(t_k)$ obeys $E[\xi_D(t)\xi_D(t')] = \delta(t-t')\eta q_e^2 \dot{N}/\Delta$, with $\Delta^{-1}$ quantifying the effective noise-bandwidth of each observation.

**Spin dynamics.** We model the dynamics of the average bulk spin of our hot atomic vapor in the SERF regime,[5,29,38] and in the presence of a magnetic field **B** in the [1,1,1] direction, i.e. $\mathbf{B} = B(\hat{x} + \hat{y} + \hat{z})/\sqrt{3}$, as

$$d\mathcal{F} = -A\mathcal{F}dt, \qquad (20)$$

where the matrix A includes dynamics due to Larmor precession and spin relaxation. It can be expressed as $A_{ij} = -\gamma B_h \varepsilon_{hij} + \Gamma_{ij}$, where $h, i, j = x, y, z$. The relaxation matrix $\Gamma$ has eigenvalues $T_1^{-1}$ and $T_2^{-1} = T_1^{-1} + T_{SE}^{-1}$ for spin components parallel and transverse to **B**, respectively. We note that in the SERF regime the decoherence introduced by SE collisions between alkali atoms is quantified by Eq. (4)[5,29].

To account for fluctuations due to spin noise in Eq. (20) we add a stochastic term $\sqrt{\sigma}d\mathbf{W}$ where $dW_h$, $h \in \{x, y, z\}$ are independent Wiener increments. Thus the statistical model for spin dynamics reads

$$d\mathcal{F} = -A\mathcal{F}dt + \sqrt{\sigma}d\mathbf{W} \qquad (21)$$

where the strength of the noise source $\sigma$, the matrix A, and the covariance matrix in statistical equilibrium $Q = E\left[\mathcal{F}(t)\mathcal{F}(t)^T\right]$ are related by the fluctuation-dissipation theorem

$$AQ + QA^T = \sigma, \qquad (22)$$

from which we obtain $\sigma = 2\Gamma Q$.

**Kalman filter.** Kalman filtering is a signal recovery method that provides continuously-updated estimates of all physical variables of a stochastic model,

along with uncertainties for those estimates. For linear dynamical systems with gaussian noise inputs, e.g. the spin dynamics of Eq. 3 with the readout of Eq. 5, the Kalman filter estimates are optimal in a least-squares sense. The KF estimates, e.g. those shown in Fig. 1b and c, indicate our evolving uncertainty about the values of the physical quantities, e.g. $\mathcal{F}_z$. As such, they provide an upper bound on the intrinsic uncertainty of these same quantities due to, e.g. quantum noise. As information accumulates, the uncertainty bounds on $\mathcal{F}_x$, $\mathcal{F}_y$ and $\mathcal{F}_z$ contract toward zero, implying the production of squeezing and entanglement. This is measurement-induced, rather than dynamically-generated entanglement. The measured signal, i.e. the optical polarization rotation, indicates a joint atomic observable: the sum of the spin projections of many atoms. For an unpolarized state such as we use here the physical back-action—which consists of small random rotations about the $\mathcal{F}_z$ axis induced by quantum fluctuations in the ellipticity of the probe—has a negligible effect.

We construct the estimator $\widetilde{\mathcal{F}}_t$ of the macroscopic spin vector using the continuous-discrete version of Kalman filtering[14]. This framework relies on a two-step procedure to construct the estimate $\tilde{\mathbf{x}}_t$, and its error covariance matrix $\boldsymbol{\Sigma} = E\left[(\mathbf{x}_t - \tilde{\mathbf{x}}_t)(\mathbf{x}_t - \tilde{\mathbf{x}}_t)^T\right]$, of the state $\mathbf{x}_t$ of a continuous-time linear-Gaussian process, in our case $\mathcal{F}(t)$, that is observed at discrete-time intervals $\Delta = t_k - t_{k-1}$. Measurement outcomes are described by the observations vector $\mathbf{z_k}$, in our case the scalar $I_k$, which is assumed to be linearly related to $\mathbf{x_t}$ via the coupling matrix $\mathbf{H_k}$ and to experience independent stochastic Gaussian noise as described previously[14].

In the first step of the Kalman filtering framework, also called the prediction step, the values at $t = t_k$, $\widetilde{\mathcal{F}}_{k|k-1}$ and $\Sigma_{k|k-1}$, are predicted conditioned on the process dynamics and the previous instance, $\widetilde{\mathcal{F}}_{k-1|k-1}$ and $\Sigma_{k-1|k-1}$, as follows:

$$\widetilde{\mathcal{F}}_{k|k-1} = \Phi_{k,k-1}\widetilde{\mathcal{F}}_{k-1|k-1}, \qquad (23)$$

$$\Sigma_{k|k-1} = \Phi_{k,k-1}\Sigma_{k-1|k-1}\Phi_{k,k-1}^T + Q_k^{\Delta}, \qquad (24)$$

where

$$\Phi_{k,k-1} = \frac{1}{3}e^{-\frac{t}{T_1}} + \frac{1}{3}e^{-\frac{t}{T_2}}$$

$$\begin{pmatrix} 2\cos(\omega_L\Delta) & -\cos(\omega_L\Delta) - \sqrt{3}\sin(\omega_L\Delta) & -\cos(\omega_L\Delta) + \sqrt{3}\sin(\omega_L\Delta) \\ -\cos(\omega_L\Delta) + \sqrt{3}\sin(\omega_L\Delta) & 2\cos(\omega_L\Delta) & -\cos(\omega_L\Delta) - \sqrt{3}\sin(\omega_L\Delta) \\ -\cos(\omega_L\Delta) - \sqrt{3}\sin(\omega_L\Delta) & -\cos(\omega_L\Delta) + \sqrt{3}\sin(\omega_L\Delta) & 2\cos(\omega_L\Delta) \end{pmatrix}$$

$$\qquad (25)$$

is the state transition matrix describing the evolution of the dynamical model Eq. (20) within the time interval $\Delta$, and

$$Q^{\Delta} = \frac{N_A}{2}\left(1 - e^{-\frac{2\Delta}{T_1}}\right) + \frac{N_A}{2}\left(1 - e^{-\frac{2\Delta}{T_2}}\right)\begin{pmatrix} 2 & -1 & -1 \\ -1 & 2 & -1 \\ -1 & -1 & 2 \end{pmatrix} \qquad (26)$$

is then the effective covariance matrix of the system noise[14].

In the second step, or update step, the information gathered through the fresh photocurrent observation $I_k$ is incorporated into the estimate:

$$\widetilde{\mathcal{F}}_{k|k} = \widetilde{\mathcal{F}}_{k|k-1} + \mathbf{K}_k\left(I_k - \mathbf{H}_k\widetilde{\mathcal{F}}_{k|k-1}\right) \qquad (27)$$

$$\boldsymbol{\Sigma}_{k|k} = (1 - \mathbf{K}_k\mathbf{H}_k)\boldsymbol{\Sigma}_{k-1|k}, \qquad (28)$$

where $\mathbf{H}_k = \left[\eta g\dot{N}, 0, 0\right]$ and the Kalman gain $\mathbf{K}_k$ is defined as

$$\mathbf{K}_k = \boldsymbol{\Sigma}_{k|k-1}\mathbf{H}_k^T\left(\mathbf{R}^{\Delta} + \mathbf{H}_k\boldsymbol{\Sigma}_{k|k-1}\mathbf{H}_k^T\right)^{-1} \qquad (29)$$

with sensor covariance $\mathbf{R}^{\Delta} = R/\Delta$ dictated by the power-spectral-density, R, of the photocurrent noise, i.e. due to photon shot-noise, and the sampling period, $\Delta$. As discussed in previous work[14] the KF is initialised according to a distribution that represents our prior knowledge about the system at time $t = t_0$ and fixes $\widetilde{\mathcal{F}}_{0|0} \sim \mathcal{N}(\boldsymbol{\mu}_0, \boldsymbol{\Sigma}_0)$, where $\boldsymbol{\mu}_0$, and $\boldsymbol{\Sigma}_0$ are the mean value and total variance of the observed data. After initialization KF estimates for the covariance matrix $\boldsymbol{\Sigma}_{k|k}$ undergo a transient and once this transient has decayed they converge to a steady state value $\boldsymbol{\Sigma}_{ss}$.

In Fig. 6 we observe this behavior for the total variance $|\Delta\mathcal{F}|^2 \equiv \text{Tr}(\Gamma_{\mathcal{F}})$ as a function of time $t = t_k$, where $\Gamma_{\mathcal{F}} = \boldsymbol{\Sigma}_{k|k}$. After about 0.8 ms, the total variance reaches steady state value which is used to compare with SQL and indicates squeezing degree. Figure 7 shows squeezing degree at different probe power, and presents the optimal probe power we observed is 2 mW.

**Validation.** To validate the sensor model we employ three validation techniques sensitive to both the statistics of the optical readout and spin noise. First, we analyse the statistics of the sensor output innovation, i.e., the difference between observations $I_k$ (data) and Kalman estimates ($\tilde{y}_k = I_k - \mathbf{H}_k\widetilde{\mathcal{F}}_{k|k-1}$). In Fig. 8, we show the $\tilde{y}_k$ histogram with the sensor output estimation error, which is described by zero-mean Gaussian process with variance equal to $\mathbf{R}^{\Delta} + \mathbf{H}_k\boldsymbol{\Sigma}_{k|k-1}\mathbf{H}_k^T$. We find 94% of $\tilde{y}_k$ data lie within a two-sided 95% confidence region of the expected

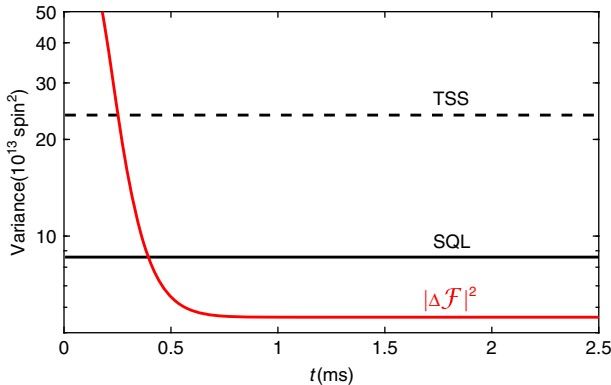

**Fig. 6 Spin variance versus Kalman filter (KF) tracking time.** The dashed black line and solid-black line are thermal spin state (TSS) noise level and standard quantum limit (SQL) for total spin. Probe power = 2 mW, $\nu_L = 1.3$ kHz.

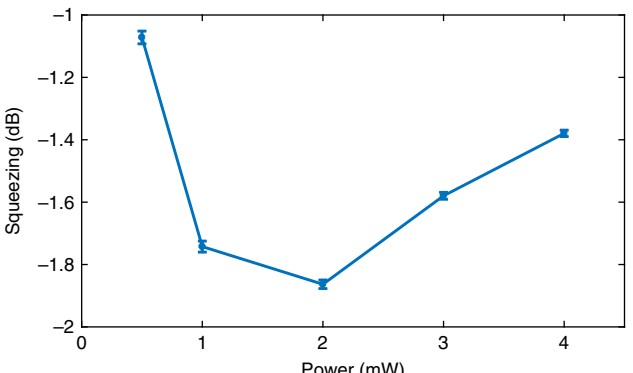

**Fig. 7 Degree of squeezing versus probe power.** We observe a squeezing optimum as 2 mW. $\nu_L = 1.3$ kHz. Error bars show ±1σ uncertainty in the variance, originating in the uncertainty of the atomic number.

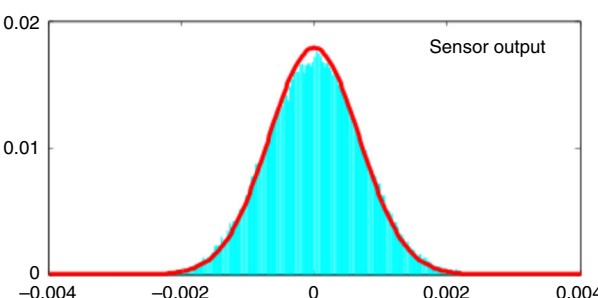

**Fig. 8 Kalman model validation based on sensor output.** Histogram (cyan) of observed sensor output innovation and Kalman estimates collected over a period of 0.35 s, compared to the zero-mean Gaussian distribution with computed sensor output estimation error (red lines). Probe power = 2 mW, $\nu_L = 1.3$ kHz.

Gaussian distribution, thus indicating a very close agreement of the model and observed statistics. We note that while being a standard technique in the validation of Kalman filtering[39], this technique for our experimental conditions is more sensitive to photon shot noise than to spin noise. Therefore, to further validate our estimates we also include two other validation techniques, designed to be sensitive to the atomic statistics on a range of time-scales.

Particularly, we perform Monte Carlo simulations based on the model described by Eqs. (2), (3) and (17) and fed with the operating conditions of our experiments and compare the power spectral density (PSD) of the simulated sensor output (Simulation) to the observed PSD of the measurements (Data), as shown in Fig. 9a. The observed agreement between Data and Simulation suggests the validity of the statistics of the spin dynamics model.

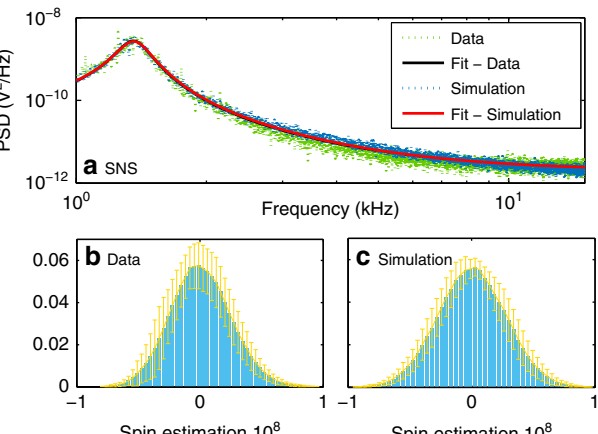

**Fig. 9 Kalman model validation based on real data (Data) and simulated data (Simulation). a** Spin noise spectroscopy (SNS) of Data (blue dots) and Simulation (green dots). The Lorentz fittings of Data (black line) and Simulation (red line) are totally overlapped. The spin distributions (see text) from Data (**b**) and Simulation (**c**) are shown as histograms. Error bars indicate plus/minus one standard deviation of histograms of 20 traces. Probe power = 2 mW, $\nu_L = 1.3$ kHz.

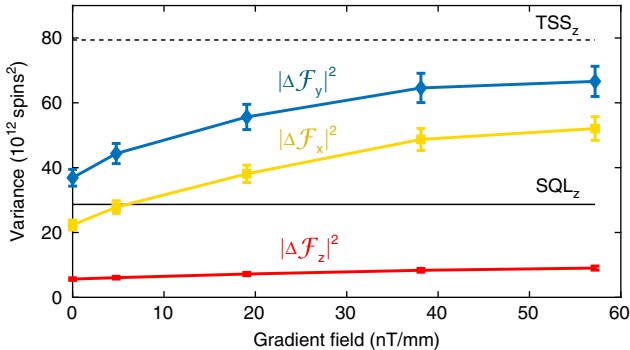

**Fig. 10 Reduction of squeezing with increasing magnetic field.** Spin variance components increase with gradient field strength. Red dots, yellow squares, blue diamonds represent $\langle \Delta \mathcal{F}_z^2 \rangle$, $\langle \Delta \mathcal{F}_x^2 \rangle$, and $\langle \Delta \mathcal{F}_y^2 \rangle$, respectively. Error bars show ±4σ uncertainty in the variance, originating in the uncertainty of the atomic number. The dashed line and solid line show thermal spin state (TSS) (7.99(18)×10¹³spins²) and standard quantum limit (SQL) (2.88(7) ×10¹³spins²) noise levels, respectively for one spin component $\mathcal{F}_h$, $h \in \{x, y, z\}$.

Finally, we employ the Kalman filter to identify the evolution of the atomic state variables based on the Simulation. We can then compare the distribution of Kalman spin-estimates from the Data versus that from Simulation. The results are shown in Fig. 9b and c, respectively. The similarity in the statistics of these two spin estimates validates the spin dynamics model. Together with the above validations, it provides a full validation of both the optical and spin parts of the model.

**Gradient field tests.** A weak gradient magnetic field is applied along the probe (z) direction by coils implemented inside the magnetic shields. In Fig. 10 we plot the three components of $|\Delta \mathcal{F}|^2$: $|\Delta \mathcal{F}_z|^2 \equiv \Gamma_{\mathcal{F}}(1, 1)$, $|\Delta \mathcal{F}_x|^2 \equiv \Gamma_{\mathcal{F}}(2, 2)$, and $\left|\Delta \mathcal{F}_y\right|^2 \equiv \Gamma_{\mathcal{F}}(3, 3)$, as a function of gradient field. Here $\Gamma_{\mathcal{F}}(i, i) = \Sigma_{ss}(i, i)$. We observe that the variance of each component increases towards the TSS noise level with gradient field. We note that due to the bias field along the [1,1,1] direction, the current ($t = t_k$) sensor reading indicates $\mathcal{F}_z$ at that time, while $\mathcal{F}_x$ and $\mathcal{F}_y$ describe components that were measured 1/3 and 2/3 Larmor cycles earlier, respectively. The combined variance is used to compute $|\Delta \mathcal{F}_z|^2$, as in Fig. 3. We note that the Stern–Gerlach (SG) effect, in which a gradient causes wave-functions components to separate in accordance with their magnetic quantum numbers, also contributes to the loss of coherence. The SG contribution is negligible, however, due to the weak gradients used here and the rapid randomization of momentum caused by the buffer gas.

## Data availability

The data that support the findings of this study are available from the corresponding author upon reasonable request. Open-access datasets from this work available at https://doi.org/10.5281/zenodo.3694692.

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

## Acknowledgements

We thank Jan Kolodynski for helpful discussions. This project has received funding from the European Union's Horizon2020 research and innovation programme under the Marie Skłodowska-Curie grant agreements QUTEMAG (no. 654339). The work was also supported by ICFOnest + Marie Skłodowska-Curie Cofund (FP7-PEOPLE-2013-COFUND), the National Natural Science Foundation of China (NSFC) (grant no. 11935012), and ITN ZULF-NMR (766402); the European Research Council (ERC) projects AQUMET (280169), ERIDIAN (713682); European Union projects QUIC (Grant Agreement no. 641122) and FET Innovation Launchpad UVALITH (800901); Quantum Technology Flagship projects MACQSIMAL (820393) and QRANGE (820405); 17FUN03-USOQS, which has received funding from the EMPIR programme co-financed by the Participating States and from the European Union's Horizon 2020 research and innovation programme; the Spanish MINECO projects MAQRO (Ref. FIS2015-68039-P), XPLICA (FIS2014-62181-EXP), OCARINA (Grant Ref. PGC2018-097056-B-I00) and Q-CLOCKS (PCI2018-092973), MCPA (FIS2015-67161-P), the Severo Ochoa programme (SEV-2015-0522); Agència de Gestió d'Ajuts Universitaris i de Recerca (AGAUR) project (2017-SGR-1354); Fundació Privada Cellex and Generalitat de Catalunya (CERCA program, QuantumCAT), the EU COST Action CA15220 and QuantERA CEBBEC, the Basque Government (Project No. IT986-16), and the National Research, Development and Innovation Office NKFIH (Contract No. K124351, KH129601). We thank the Humboldt Foundation for a Bessel Research Award.

## Author contributions

J.K. and M.W.M. designed the experiment, analyzed the data and wrote the paper. J.K., C.T., and V.G.L. performed the experiment. J.K., R.J.-M., and M.W.M built the Kalman filter model. G.T. and M.W.M. built the entanglement witness. M.W.M. supervised the project.

## Competing interests

The authors declare no competing interests.
