## [Peer Review File · Nature Communications]

Reviewers' Comments:

Reviewer #1:

Remarks to the Author:

Kong et al. present a solid work about generating a large entangled state in a hot atomic vapor. Even though I think this manuscript fulfills the criteria of Nature Communication for novelty and interest for a broader community, it requires a revision before publishing.

Below I list my comments (not in the order of importance)

1) sometimes the logical flow of the story breaks:

a) in the introductory section, they mention the notion "[1,1,1]" for a magnetic field direction which is not explained right away. It becomes clear only after looking at the fig 1 at the next page. At least, authors should add a reference to this figure

b) in the Result - material system section, authors mention that at Larmor freq < 5 kHz the desired regime of SERF is achieved. Only a half page later they explain why it is so.

c) connected to b) I believe, authors should give a brief explanation of physics behind SERF regime in the introductory paragraph. This will help non-experts better understand this work without looking up references

2) a bit more references are needed to cover the statements made by authors.

a) at the beginning, "the same processes also decouple ... which increase the spin coherence time"

b) at the Discussion, "... observed macroscopic singlet states shares several traits with a spin liquid state,.."

3) Regarding the generated entangled state:

a) I did not find that the authors specified which degree of freedom is entangled (electron or nuclear spin)

b) is it possible to write down the form of the state or to describe it somehow

c) what is the fidelity of the entangled state? In other words, is it useful for sensing and other applications mentioned in the introduction?

4) Kalman filter. The authors discuss this filtering in length in the Methods. I am actually curious, are there any other techniques to analyze their data giving that data in Fig 1b have quite good SNR?

5) The authors mention the optimal optical power of 2 mW, it would be interesting to see an explanation why this is the case

6) one of the main applications stated in the introduction is (magnetic) sensing. It seems from Fig 1 that the vapor cell is enclosed in magnetic shields. Will this hamper sensing of small external magnetic fields?

Reviewer #2:

Remarks to the Author:

Summary of reported work

The authors present measurements of spin noise in a hot rubidium vapor. The discretized measurements of spin noise are fed into a Kalman filter which, given the spin noise dynamics, estimates the mean value of the noise and its fluctuations. The authors use spin squeezing inequalities to claim that the measured fluctuations around the mean of the polarimeter signal, which reflects the collective spin along the laser beam axis, are smaller than the standard quantum limit (and the thermal state limit). The authors claim that for this to happen, a macroscopically large number of atoms must have been entangled (due to the measurement performed by the light field), in particular, into a macroscopic singlet state. To support their case, the authors present further systematic checks, like squeezing dependence on light power and magnetic gradient.

General comments

As a first comment, the manuscript is very clearly written and the work therein seems to be a thoroughly explained experiment. The figures are very nice and informative. Apart from the somewhat technical discussion on the workings of the Kalman filter, the manuscript is accessible by the general reader.

This work could be a significant step forward in the field of quantum sensing, and a testimony to the fact that great strides can still be made in atomic physics and quantum metrology with relatively simple experimental setups augmented with fresh experimental approaches and an insightful theoretical analysis unraveling the details of the underlying physics.

Technical comments

For the sake of being fully convinced, further scrutinizing the reported results, and clarifying some subtle points, I would like to better understand the following:

(1a) In Figs. 1b,c the authors compare the SQL and TSS with the error band around the running mean of the spin noise signal, as extracted by the KF. However, this error band represents the

noise of spin noise, whereas I would think that what limits a metrological measurement is not the noise on top of the noise, but the noise itself. In other words, the whole oscillating signal in Fig. 1b, being stochastic and hence unpredictable for timescales $> T_2$, is what I would think sets a limit to the measurement precision when integrating for times $> T_2$, the noise on top of the noise (the spread of the shot-to-shot measurements) being a second-order effect. By visually inspecting the rms amplitude of the spin noise signal itself with $\frac{1}{4}$ of the SQL and TSS 4σ -bars, it seems that this amplitude is quite larger.

(1b) Hence the fact the KF (given enough points and the underlying dynamics) estimates with a given precision a part of the noise signal where some randomly generated coherence dominates the dynamics does not imply that it can predict an intrinsically unpredictable noise signal for times $> T_2$. That is, the KF might allow to precisely estimate the noise amplitude in a "coherent" snapshot lasting for about T_2 , but this noise amplitude itself will be random in many such snapshots, the distribution being set by the random bursts of the spin noise signal itself (i.e. its amplitude) and not the shot-to-shot fluctuations. So it is not clear if the presented sub-SQL measurement is a "metrological sub-SQL".

(1c) To elaborate a bit more, what I understand as spin noise is the spontaneously generated collective spin (the randomly created oscillations lasting for about T_2 , then randomly regenerating themselves etc). As recently shown (PR Research 1, 033017, 2019), spin exchange collisions (as well as other kinds of binary collisions) continuously generate spin noise due to the quantum randomness of the post-collision states. Now, the so produced non-zero collective spin fluctuations (as in Fig. 1) scale with atom number as do the collective quantum uncertainties of spin observables in specific states (SQL or TSS), and could also be numerically similar. But attributing the former to the latter is to my understanding far from obvious if not plainly incorrect, as it is also far from obvious that such fluctuations (generated by binary collisions) are in any way related to the actual measurement of the collective spin by the light field. I'm not saying that the authors make such attributions, I'm just "thinking aloud" and would just like to "disentangle" three concurrent issues that I find confusing: spin noise generated by collisions, shot-to-shot variations in measured spin noise, and SQL/TSS bars.

(1d) Related to this is a statement in page 3, left column, where the authors state that it is the measurement (I add "with the light field") that reduces the pink error band in the first few microseconds of the measurement. However, one could claim that it is just the numerical inability

of a classical estimator to make an estimate with only a few points available around $t=0$. Indeed, in the distribution of the actual measurement points just after $t=0$ there is no apparent change in their spread, so there is no evidence of shrinking of the actual measurement uncertainty after the onset of measurement (i.e. due to acquisition of information).

(1e) So from the authors' perspective I understand that it is the measurement of the collective spin by the light field that projects the atoms to a non-classical state, and the pink error band reflects the atomic noise of this alleged non-classical state, much like we plot coherent states of light with a thickened sine wave, the thickness reflecting the quantum fluctuations of the electric field. This pink error band being smaller than the SQL, there is entanglement, the authors claim.

(1f) Based on the points (1a)-(1d), however, I could paint a different physical picture. Random atomic collisions generate the spin noise signal shown in Fig. 1, the atomic state being some separable state determined by the (still not well understood) physics of spin exchange collisions at the quantum noise level. This state could have even zero or some other nonzero but small variance along z , while the observed pink error band could be due to some classical noise source. The subtlety is that since the transverse components and their variances are not measured in order to assess metrological spin squeezing, the authors measure along 1D and rely on the 3D KF estimates to find the total spin variance. But, for the sake of arguing, the atomic collision dynamics (which do generate quantum fluctuations of the collective spin) coupled with the collective measurement induced by the light field interaction could lead to rather complicated dynamics not captured by Eq. 3 and the KF. Essentially, how do we really know that these fluctuations (the pink error band) are of atomic origin? Usually in studies of spin noise, the scaling of the total noise power under the spectra in Fig. 2a is plotted against atom number, and a linear scaling with atom number shows that these are indeed spin noise spectra. But now we are talking about consecutive shot-to-shot fluctuations of the measured spin noise. How do they scale with atom number? I understand it might be hard to stay in the SERF regime and significantly vary atom number, but do the authors have some other way to elaborate on this?

(2) The authors claim that a substantial fraction (30%) of the atoms are entangled in a macroscopic singlet state. The singlet being magnetically silent, I would expect that outside the SERF regime (large Larmor frequencies where purported variances approach SQL) these atoms would cease to be magnetically silent, hence the rms amplitude of the spin noise signal itself, and not just the fluctuations of the amplitude around its running mean, should grow larger.

Equivalently (since this might be hard to observe when the linewidths increase), I would expect the plotted spectra in Fig. 2a to contain different integrated noise powers, if I assume that at the low frequencies of the SERF regime a large fraction of atoms do not contribute to the average spin noise signal but only to its shot-to-shot variance. With the quality of the spectra and the fits, a 30% effect in integrated noise power should be readily observable. Is there such an effect? If not, why?

Essentially, all my questions are intertwined and boil down to the aforementioned subtle concurrence of (i) spin noise itself, (ii) shot-to-shot fluctuations thereof and (iii) theoretical uncertainty bars. I'm looking forward to the authors helping me to clarify the above.

Wording and referencing

(1) It would probably be prudent to abstain from statements on how many orders of magnitude this measurement surpasses other measurements in entanglement metrics, since not all measurements can be compared in a straightforward way, nor is such a global comparison the main objective of this work, the physics details of previous and the current measurement being rather subtle.

(2) I'm delighted to see the first experiment hinting on the resilience of entangled hot-vapor states to binary collisions, as predicted in 2008 (PRL 100, 073002). Furthermore, the paper experimentally "demonstrating coherent inter-species quantum state transfer", ahead of the theoretical predictions of Ref. 17, is PRA 90, 032705 (2014).

Minor Comments

(1) At the end of the abstract the same sentence is repeated twice.

(2) Eq. 12, the right side should be $(N_A - N_e) * F_{\alpha}$

Sincerely,

Iannis Kominis

Department of Physics

University of Crete

Reviewer #3:

Remarks to the Author:

This is a very interesting paper. The number of particles that are reported to be entangled constitutes a new record. The regime of operation of the experiment is also remarkable. It is fascinating to think about the fact that the particles participating in the entanglement are constantly changing because of the spin-exchange collisions, but that the entanglement itself is preserved. I do think that the paper deserves to be published in Nature Communications, but I would like the authors to address my questions below. I am listing them roughly in order of importance, starting with the most important.

The authors give a bound for the number of entangled particles. I am wondering whether it would be possible to also infer something about the type of entanglement, i.e. two-particle singlets versus more complex multi-party entanglement. I am in particular thinking about the methods used in this paper: <https://journals.aps.org/prl/abstract/10.1103/PhysRevLett.86.4431> Could they be adapted to the present experiment? Or is it clear that no multi-party entanglement will be created?

The authors also estimate the spatial range of entanglement based on applying a magnetic field gradient. They find a range that is much greater than a wavelength, but smaller than the size of the cell. Can this result be understood quantitatively?

I gained some understanding of the SERF regime from reading this paper, but I do feel that the explanation could still have been more comprehensive and self-contained. Some related detail questions include the size of A_{hf} (it would have been nice to see that somewhere on page 2 or 3), and the meaning of the 'nuclear slowing-down factor' q .

Finally a minor point, there is a lot of repetition towards the end of the abstract.

Reply to comments of Reviewer 1

We thank Reviewer 1 for a careful and detailed reading of the manuscript, and for identifying several aspects that required improvement. Below we give a point-by-point response to the Reviewer's comments. Resulting changes are indicated **in blue** in the revised manuscript.

Comment 1. sometimes the logical flow of the story breaks:

a) in the introductory section, they mention the notion "[1,1,1]" for a magnetic field direction which is not explained right away. It becomes clear only after looking at the fig 1 at the next page. At least, authors should add a reference to this figure.

b) in the Result - material system section, authors mention that at Larmor freq < 5 kHz the desired regime of SEFR is achieved. Only a half page later they explain why it is so.

c) connected to b) I believe, authors should give a brief explanation of physics behind SERF regime in the introductory paragraph. This will help non-experts better understand this work without looking up references

Response: We thank the Reviewer for these suggestions. We have added the corresponding reference and explanation according to the Reviewer's suggestions.

Comment 2. a bit more references are needed to cover the statements made by authors.

a) at the beginning, "the same processes also decouple ... which increase the spin coherence time"

b) at the Discussion, "... observed macroscopic singlet states shares several traits with a spin liquid state,..."

Response: We thank the Reviewer for these comments. We have added the corresponding references according to the Reviewer's comments.

Comment 3. Regarding the generated entangled state:

a) I did not find that the authors specified which degree of freedom is entangled (electron or nuclear spin)

b) is it possible to write down the form of the state or to describe it somehow

c) what is the fidelity of the entangled state? In other words, is it useful for sensing and other applications mentioned in the introduction?

Response: We thank the Reviewer for these comments. Indeed, we did not specify which degree of freedom is entangled. Spin squeezing theory allows us to say that the atoms are entangled, but (to date at least) does not indicate in what degree of freedom. We can hypothesize that the entanglement follows the chain of interactions: The optical probe interacts with the electron spin and orbital angular momentum, which presumably entangles the electron spins of different atoms, including atoms at a distance, because the probe light interacts with all of the atoms. The electron spin is coherently coupled to the nuclear spin by the hyperfine interaction, so we can expect entanglement also of the nuclear spins of distant atoms. Finally, collisions between atoms exchange their electron spins. We can hypothesize thus entangled states involving the electrons and nuclei of clusters of atoms in one place, with clusters of atoms in another. Fortunately, the squeezing is useful for metrological purposes even if we don't know the exact nature

of the entangled state. One example that has been studied in detail is the magnetic gradiometer¹.

Comment 4. Kalman filter. The authors discuss this filtering in length in the Methods. I am actually curious, are there any other techniques to analyze their data given that data in Fig 1b have quite good SNR?

Response: As the Referee suggests, there are indeed other ways to analyze time-domain data. Probably the most commonly used method is to compute the conditional variance of fits to the data². We have previously used this method for spin squeezing using a sequence of probe pulses^{3,4}. This conditional variance method has several intrinsic drawbacks when working with diffusive continuous-time data: No simple parametrized fit function captures accurately the diffusion process over long time scales; using non-simple fit functions makes the fits less accurate (the so-called "bias-variance tradeoff"); and any simple fit function (e.g. a sinusoid with amplitude and phase that is polynomial in time) has a nonlinear dependence on its parameters (e.g. F_z is a sinusoidal, not linear, function of the phase). The conditional variance approach is also simply cumbersome, in that it requires a large amount of data to be fit for every point in the time series. In contrast, the Kalman filter based on Eq. (3) is efficient, uniquely defined, linear, and optimal in a least-squares sense. We appreciate that the Kalman filter approach is relatively new and unfamiliar in atomic sensing, but it has been shown to be remarkably accurate in describing the statistical properties of spin noise. See for example Jimenez-Martinez et al.⁵, an experiment we performed precisely to check the accuracy of the Kalman filter in this context.

Comment 5. The authors mention the optimal optical power of 2 mW, it would be interesting to see an explanation why this is the case

Response: We thank the Reviewer for this good question. There are many parameters that could affect the entanglement generation, and the optical power is one of them. Higher optical power will make stronger interaction with atoms therefore in principle will bring us better signal-to-noise ratio, however at the same time, it will increase the power broadening of the spin noise resonance, which is to say it will accelerate the spin relaxation and diffusion. In optimizing the experiment we acquired data with different probe powers, keeping other parameters fixed and found 2 mW to be optimal for spin squeezing.

Comment 6. one of the main applications stated in the introduction is (magnetic) sensing. It seems from Fig 1 that the vapor cell is enclosed in magnetic shields. Will this hamper sensing of small external magnetic fields?

Response: We thank the Reviewer for this question. The reviewer is correct, our vapor cell is enclosed in a 4-layer magnetic shield which prevents outside fields from reaching the sensor. This is the usual configuration for testing magnetic sensors, because it blocks environmental noise. The configuration is also used for precision sensing when the source is small enough to be placed inside the shields along with the sensor. For measuring the field from larger sources, e.g. in human brain magnetic field measurements, the source and sensor are placed together in a magnetically shielded room, which is simply a larger magnetic shield. We note that shielding is especially important for SERF magnetometers, which are extremely sensitive (sub-fT/ $\sqrt{\text{Hz}}$), but only have this sensitivity when the total field strength is small. A different class of magnetometers is used for unshielded measurements, e.g. measurements of the earth field.

1. I. Urizar-Lanz, P. Hyllus, I. L. Egusquiza, M. W. Mitchell, and G. Tóth, "Macroscopic singlet states for gradient magnetometry," *Phys. Rev. A* **88**, 013626 (2013).
2. G. Vasilakis, V. Shah, and M. V. Romalis, "Stroboscopic backaction evasion in a dense alkali-metal vapor," *Phys. Rev. Lett.* **106**, 143601 (2011).
3. N. Behbood, F. Martin Ciurana, G. Colangelo, M. Napolitano, G. Tóth, R. J. Sewell, and M. W. Mitchell, "Generation of macroscopic singlet states in a cold atomic ensemble," *Phys. Rev. Lett.* **113**, 093601 (2014).
4. G. Colangelo, F. M. Ciurana, L. C. Bianchet, R. J. Sewell, and M. W. Mitchell, "Simultaneous tracking of spin angle and amplitude beyond classical limits," *Nature* **543**, 525 (2017).
5. R. Jiménez-Martínez, J. Kołodyński, C. Troullinou, V. G. Lucivero, J. Kong, and M. W. Mitchell, "Signal tracking beyond the time resolution of an atomic sensor by Kalman filtering," *Phys. Rev. Lett.* **120**, 040503 (2018)

Reply to Reviewer 2

We thank Reviewer 2 for the very in-depth questions and comments. Below we give a bit of context to frame the discussion and then reply to the Reviewer's queries. Resulting changes are marked in blue in the revised manuscript.

Context

As the Reviewer is probably quite aware, SERF regime vapors have a complex physics, and there is to date no theory that can accurately describe non-classical states in these vapors, nor accurately describe the quantum effects of non-destructive measurement, e.g. Faraday rotation probing. One might expect that SERF-regime vapors would be very good for quantum-enhanced sensing, because they combine high optical depth (which in simpler systems makes for good QND measurements and good measurement-induced squeezing) with long coherence times (which makes for good sensitivity). Or one might expect that SERF-regime vapors are very *bad* for quantum enhanced sensing, because the SERF physics would scramble any entangled/squeezed states through the fast and random spin-exchange. We saw an opportunity to test this latter hypothesis, by trying to make a singlet state. Our approach is at heart the same as we used in Behbood et al PRL 2014 [1] where we used cold atoms and pulsed measurements, with a (1,1,1) B-field and a 1/3 period wait time between measurements, to get statistics of all three components of \mathcal{F} . For the SERF experiment we use continuous measurements and thus the Kalman filter is the most appropriate analysis tool.

A natural question is “why use a non-polarized state rather than a polarized state?” Of course, we are ultimately interested in polarized states, because these are relevant to sensing. But if the question is whether SERF physics can support entanglement/squeezing, making an unpolarized/singlet state is arguably a more stringent test, because strongly polarized states can be protected against spin-exchange relaxation by other mechanisms. The singlet is also much less sensitive to magnetic and technical noise. Finally, for an unpolarized ensemble the statistical model is linear, allowing us to use an ordinary Kalman filter rather than an extended Kalman filter. If we had included optical pumping in the experiment, and tried to create a strongly-polarized state, the statistical model would have to be nonlinear to account for the saturation of polarization due to optical pumping.

Response to points raised

Comment 1a. (1a) In Figs. 1b, c the authors compare the SQL and TSS with the error band around the running mean of the spin noise signal, as extracted by the KF. However, this error band represents the noise of spin noise, whereas I would think that what limits a metrological measurement is not the noise on top of the noise, but the noise itself. In other words, the whole oscillating signal in Fig. 1b, being stochastic and hence unpredictable for timescales $> T_2$, is what I would think sets a limit to the measurement precision when integrating for times $> T_2$, the noise on top of the noise (the spread of the shot-to-shot measurements)

being a second-order effect. By visually inspecting the rms amplitude of the spin noise signal itself with $\frac{1}{4}$ of the SQL and TSS 4σ -bars, it seems that this amplitude is quite larger.

Response: As the Reviewer writes, one may reasonably expect that measurement of slow signals, i.e. of frequency components larger than $1/T_2$, will be limited by spin diffusion, not by the instantaneous uncertainty of the state. Allowing that this is the case, there is nonetheless the possibility to improve the measurement of faster frequency components. This question is studied in Shah et al. PRL 2010 [2], with which the Reviewer is presumably familiar, so we won't elaborate on it now. Assuming the conclusions of that work are correct, there are signals (fast ones) for which the uncertainty in the Kalman filter estimates (what the Reviewer calls "noise on top of the noise") would be the limiting factor. See also Jimenez-Martinez et al. PRL 2018 [3]. At the same time, we remind the Reviewer that our goal here was to test whether SERF regime vapors can support squeezed/entangled states, which may have relevance to measurements beyond spin noise spectroscopy. The relevant time scale for this test is the spin-thermalization time scale, which is much shorter than T_2 .

Comment 1b. (1b) Hence the fact the KF (given enough points and the underlying dynamics) estimates with a given precision a part of the noise signal where some randomly generated coherence dominates the dynamics does not imply that it can predict an intrinsically unpredictable noise signal for times $> T_2$. That is, the KF might allow to precisely estimate the noise amplitude in a "coherent" snapshot lasting for about T_2 , but this noise amplitude itself will be random in many such snapshots, the distribution being set by the random bursts of the spin noise signal itself (i.e. its amplitude) and not the shot-to-shot fluctuations. So it is not clear if the presented sub-SQL measurement is a "metrological sub-SQL".

Response: As described in our response to (1a) we agree with the highlighted statements. The question of the metrological value will of course depend on what one aims to measure. If the spin noise itself is of interest, then a low-noise, non-destructive readout will more clearly reveal the spin noise, while also not perturbing it. This has some metrological value, see for example these publications by Lucivero et al. on the topic [4, 5]. Nonetheless, externally imposed *changes* in the spin state will probably be more often of interest than the spin noise itself. For example, in a FID magnetometer, the rate of precession indicates the instantaneous field. In this scenario the changes of interest may well be in the sub- T_2 time scale, and the precision of the instantaneous estimates would be very relevant.

Comment 1c. (1c) To elaborate a bit more, what I understand as spin noise is the spontaneously generated collective spin (the randomly created oscillations lasting for about T_2 , then randomly regenerating themselves etc). As recently

shown (PR Research 1, 033017, 2019), spin exchange collisions (as well as other kinds of binary collisions) continuously generate spin noise due to the quantum randomness of the post-collision states. Now, the so produced non-zero collective spin fluctuations (as in Fig. 1) scale with atom number as do the collective quantum uncertainties of spin observables in specific states (SQL or TSS), and could also be numerically similar. But attributing the former to the latter is to my understanding far from obvious if not plainly incorrect, as it is also far from obvious that such fluctuations (generated by binary collisions) are in any way related to the actual measurement of the collective spin by the light field. I'm not saying that the authors make such attributions, I'm just "thinking aloud" and would just like to "disentangle" three concurrent issues that I find confusing: spin noise generated by collisions, shot-to-shot variations in measured spin noise, and SQL/TSS bars.

Response: Regarding the green-highlighted text. This is probably well known by the Reviewer, but it bears repeating: SE collisions, in combination with the HF interaction, cause a collection of atoms to relax toward a spin thermal state. In this process, the net spin is unchanged, because both SE and HF processes conserve total angular momentum. Because the spin-thermal state is the highest entropy state with a given net spin angular momentum, this adds noise to any state that is not already a spin-thermal state. Regarding the yellow-highlighted text: For the spin noise shown in Fig. 1, the rate of relaxation to the spin thermal state is much faster than either the spin precession or the spin diffusion. Because of this, the observed collective spin oscillation and fluctuations cannot be due to the SE process, but rather to processes that modify the total angular momentum, which include binary spin destruction (SD) collisions, diffusion of atoms into and out of the probed region, and scattering of probe light. Regarding the blue-highlighted text: The measurement interaction causes three relevant "measurement back action" effects on the state of the atoms. The first two we will call "dynamical effects" because they change the observable \mathcal{F} via Hamiltonian. These are 1) spin rotation caused by the ellipticity of the probe light, which produces an optical Zeeman shift. Because the probe is linearly polarized, it only has a nonzero ellipticity through quantum fluctuations, so the generated rotation is random with zero mean. 2) scattering of the probe light, which also makes a random contribution to the spin. The last effect 3) we will call an "information effect" because it is caused by projecting the quantum state in the act of measurement, not directly by the dynamics. The probe gives information about the state, reducing our uncertainty about \mathcal{F}_z . This projects the state into a state more closely resembling an eigenstate of the \mathcal{F}_z operator.

Note that for an unpolarized state like the one used here, effect 1) is negligible in practice. It rotates the state by a small angle, causing $\mathcal{F}_x \rightarrow \mathcal{F}_x \cos\theta + \mathcal{F}_y \sin\theta$ and similar for \mathcal{F}_y , where θ is the spin rotation angle produced by the optical Zeeman shift. Because $\mathcal{F}_x \sim \mathcal{F}_y \sim 1/\sqrt{N}$, the $\sin\theta$ term can be neglected,

provided $|\theta| \ll 1$. In contrast, for a state polarized along the y direction, such that $\mathcal{F}_x \sim 1/\sqrt{N}$, $\mathcal{F}_y \sim N$, this condition would be $|\theta| \ll 1/N^{3/2}$. The measurement-induced spin rotation is always a significant effect on the state when the measurement produces an uncertainty comparable to the SQL.

Comment 1d. (1d) Related to this is a statement in page 3, left column, where the authors state that it is the measurement (I add “with the light field”) that reduces the pink error band in the first few microseconds of the measurement. However, one could claim that it is just the numerical inability of a classical estimator to make an estimate with only a few points available around $t=0$. Indeed, in the distribution of the actual measurement points just after $t=0$ there is no apparent change in their spread, so there is no evidence of shrinking of the actual measurement uncertainty after the onset of measurement (i.e. due to acquisition of information).

Response: The Reviewer seems to be looking for a physical effect on the spins due to the measurement. As described in response to (1c), the physical effect (a random spin rotation) has a small effect on the spin components of an unpolarized state like we use here. The important effect is the informational effect: *our* uncertainty about the spin state is reduced as we get more data. Because the quantum uncertainty of an observable is upper-bounded by our uncertainty about it, this allows us to detect squeezed states. This is not in principle different from other spin squeezing experiments, e.g. Koschorreck et al. PRL 2010 [6, 7] and especially Behbood et al. PRL 2014 [1].

Comment 1e. (1e) So from the authors’ perspective I understand that it is the measurement of the collective spin by the light field that projects the atoms to a non-classical state, and the pink error band reflects the atomic noise of this alleged non-classical state, much like we plot coherent states of light with a thickened sine wave, the thickness reflecting the quantum fluctuations of the electric field. This pink error band being smaller than the SQL, there is entanglement, the authors claim.

Response: The (width of the) pink band represents our uncertainty about the spin observable \mathcal{F}_z . Note that the Kalman filter provides also uncertainties for the other components, in the form of a covariance matrix that describes both the variances and the correlations of \mathcal{F}_x , \mathcal{F}_y and \mathcal{F}_z . Because it reflects our knowledge of these variables, the covariance matrix provides upper bounds on the uncertainty of the state.

Comment 1f. (1f) Based on the points (1a)-(1d), however, I could paint a different physical picture. Random atomic collisions generate the spin noise signal shown in Fig. 1, the atomic state being some separable state determined by the (still not well

understood) physics of spin exchange collisions at the quantum noise level. This state could have even zero or some other nonzero but small variance along z , while the observed pink error band could be due to some classical noise source. The subtlety is that since the transverse components and their variances are not measured in order to assess metrological spin squeezing, the authors measure along 1D and rely on the 3D KF estimates to find the total spin variance. But, for the sake of arguing, the atomic collision dynamics (which do generate quantum fluctuations of the collective spin) coupled with the collective measurement induced by the light field interaction could lead to rather complicated dynamics not captured by Eq. 3 and the KF. Essentially, how do we really know that these fluctuations (the pink error band) are of atomic origin? Usually in studies of spin noise, the scaling of the total noise power under the spectra in Fig. 2a is plotted against atom number, and a linear scaling with atom number shows that these are indeed spin noise spectra. But now we are talking about consecutive shot-to-shot fluctuations of the measured spin noise. How do they scale with atom number? I understand it might be hard to stay in the SERF regime and significantly vary atom number, but do the authors have some other way to elaborate on this?

Response: Regarding the yellow text: Random spin processes (SD collisions, diffusion) do indeed cause the spin noise signal of Fig. 1. As described in the response to (1c), the (width of the) pink band indicates our uncertainty about \mathcal{F}_z as a function of time as the measurement proceeds. The quantum uncertainty of \mathcal{F}_z cannot be larger than our uncertainty about \mathcal{F}_z , so the width of the pink band is an upper bound. It is probably worth pointing out that this uncertainty comes to an equilibrium value (see for example Fig. 6) due to a competition of diffusion (which pushes the uncertainties toward their thermal state values) and measurement, which pushes them toward zero. Regarding the blue text: We make various checks of the validity of Eq. (3) and the KF. In fact, these are equivalent because the KF is derived from Eq. (3). See the “Validation” section of the Methods. Based on these, we believe that the KF results accurately describe the dynamics of \mathcal{F} . Regarding the green text: Scaling with atom number and photon flux is often used to separate different noise contributions (atomic quantum noise, atomic technical noise, photon shot noise, etc.). One main motivation for doing this is to identify the SQL. This strategy works well in scenarios involving non-interacting particles, because each noise contribution has a simple polynomial scaling. When the particles begin to interact, this is no longer the case. This can be observed for example in spin noise spectra: outside the SERF regime the area of the spin noise peak (i.e., the integrated noise power) is proportional to atom number, and independent of Larmor frequency and relaxation rate. In the transition from the non-SERF to SERF regimes it is not proportional to atom number. For example, in Fig. 2a the spin noise peaks are not of the same area, even though the density is the same. For this reason, we did not consider it appropriate to use scaling to determine SQL. Instead, we

made a direct calibration of the number of atoms participating. This used the spin noise linewidth versus density, and the known value of the SE collision rates, as a calibration for the density (see Methods: Density Calibration), a direct measurement of the beam dimensions to determine the effective volume of the sample, and the computed rotation efficiency of the vapor (see Methods: Observed Spin Signal). The SQL of the total noise $\text{var}(\mathcal{F}_x) + \text{var}(\mathcal{F}_y) + \text{var}(\mathcal{F}_z)$ for N atoms in a thermal state is computed in Methods: Entanglement Witness.

Comment 2. (2) The authors claim that a substantial fraction (30%) of the atoms are entangled in a macroscopic singlet state. The singlet being magnetically silent, I would expect that outside the SERF regime (large Larmor frequencies where purported variances approach SQL) these atoms would cease to be magnetically silent, hence the rms amplitude of the spin noise signal itself, and not just the fluctuations of the amplitude around its running mean, should grow larger. Equivalently (since this might be hard to observe when the linewidths increase), I would expect the plotted spectra in Fig. 2a to contain different integrated noise powers, if I assume that at the low frequencies of the SERF regime a large fraction of atoms do not contribute to the average spin noise signal but only to its shot-to-shot variance. With the quality of the spectra and the fits, a 30% effect in integrated noise power should be readily observable. Is there such an effect? If not, why?

Response: As already mentioned in response to (1f), there *is* a difference in the integrated noise powers (the “area” we called it) between the SERF and non-SERF regimes. This does not have anything to do with the QND measurement or the generation of singlet, however. A simple proof of this is that if we turn down the probe power the spin squeezing goes away, but the spin noise spectra look the same (that is, the atomic contribution to the rotation angle noise is the same; the shot noise contribution to the angular noise is larger at lower probe power). How is it possible that we are putting 30% of the spins into singlets and it does not reduce the spin noise? One way to understand this is to note that by measurement we are only shrinking the uncertainty of \mathcal{F} , not its average. As seen in Fig. 1, the average continues to diffuse in the same way as it would without the measurement (we ignore power broadening, which in practice is small and in principle can be made arbitrarily small through large OD). Nonetheless, due to the measurement we know the value of \mathcal{F} with uncertainty below the SQL, and so the uncertainty of the state must also have been reduced below this level. And spin squeezing theory tells us that the only way to reduce the quantum uncertainty below the SQL is to make singlets. If this still appears contradictory, consider this scenario: we have $N = 10^{12}$ atoms experiencing spin diffusion, which causes the average spin polarization $\langle \mathcal{F} \rangle$ to wander about zero with excursions of typical magnitude $\delta \langle \mathcal{F}_z \rangle \sim \sqrt{N} = 10^6$. This condition is compatible with different states, with different uncertainties: 1) a small minority (~one part in

10^6) of the atoms are polarized, and the rest are in a thermal state. The uncertainty of \mathcal{F} is that of the thermal state (to within a few parts in 10^6) 2) a small minority (\sim one part in 10^6) of the atoms are polarized, and the rest are in singlet states. The uncertainty of \mathcal{F} is that of the singlet state state (to within a few parts in 10^6) 3) states that interpolate between 1 and 2, with corresponding uncertainty of \mathcal{F} . What the QND measurement does, apparently, is to convert non-entangled unpolarized atoms (e.g. thermal states) into entangled non-polarized atoms (singlets).

Comment 3. Essentially, all my questions are intertwined and boil down to the aforementioned subtle concurrence of (i) spin noise itself, (ii) shot-to-shot fluctuations thereof and (iii) theoretical uncertainty bars. I'm looking forward to the authors helping me to clarify the above.

Response: We thank the Reviewer for the extensive discussion of the relationship of physical and measurement statistical uncertainties. We hope that our answers have clarified some of this. To make these several points clearer in the manuscript, we have added an expository paragraph to the start of the Kalman filter section.

Comment 4. Wording and referencing

Comment 4. (1) It would probably be prudent to abstain from statements on how many orders of magnitude this measurement surpasses other measurements in entanglement metrics, since not all measurements can be compared in a straightforward way, nor is such a global comparison the main objective of this work, the physics details of previous and the current measurement being rather subtle.

Response: We thank the Reviewer for this advice. We have modified the relevant passage on page 3 to include relevant details of the cited experiments.

Comment 4. (2) I'm delighted to see the first experiment hinting on the resilience of entangled hot-vapor states to binary collisions, as predicted in 2008 (PRL 100, 073002). Furthermore, the paper experimentally "demonstrating coherent inter-species quantum state transfer", ahead of the theoretical predictions of Ref. 17, is PRA 90, 032705 (2014).

Response: We have added these references to the paper as reference 11 and 19.

Comment 5. Minor Comments

- (1) At the end of the abstract the same sentence is repeated twice.
- (2) Eq. 12, the right side should be $(N_A - N_e) * F_{\alpha}$

Response: We thank the Reviewer for the suggested corrections. We have implemented them.

References

- [1] N. Behbood, F. Martin Ciurana, G. Colangelo, M. Napolitano, G. Tóth, R. J. Sewell, and M. W. Mitchell, "Generation of Macroscopic Singlet States in a Cold Atomic Ensemble," *Phys. Rev. Lett.* **113**, 093601 (2014). <http://link.aps.org/doi/10.1103/PhysRevLett.113.093601>
- [2] V. Shah, G. Vasilakis, and M. V. Romalis, "High bandwidth atomic magnetometry with continuous quantum nondemolition measurements," *Phys Rev Lett* **104**, 013601 (2010). <http://dx.doi.org/10.1103/PhysRevLett.104.013601>
- [3] R. Jiménez-Martínez, J. Kołodziej, C. Troullinou, V. G. Lucivero, J. Kong, and M. W. Mitchell, "Signal Tracking Beyond the Time Resolution of an Atomic Sensor by Kalman Filtering," *Phys. Rev. Lett.* **120**, 040503 (2018). <https://link.aps.org/doi/10.1103/PhysRevLett.120.040503>
- [4] V. G. Lucivero, A. Dimic, J. Kong, R. Jiménez-Martínez, and M. W. Mitchell, "Sensitivity, quantum limits, and quantum enhancement of noise spectroscopies," *Phys. Rev. A* **95**, 041803 (2017). <https://link.aps.org/doi/10.1103/PhysRevA.95.041803>
- [5] V. G. Lucivero, R. Jiménez-Martínez, J. Kong, and M. W. Mitchell, "Squeezed-light spin noise spectroscopy," *Phys. Rev. A* **93**, 053802 (2016). <http://link.aps.org/doi/10.1103/PhysRevA.93.053802>
- [6] M. Koschorreck, M. Napolitano, B. Dubost, and M. W. Mitchell, "Sub-Projection-Noise Sensitivity in Broadband Atomic Magnetometry," *Phys. Rev. Lett.* **104**, 093602 (2010). <http://link.aps.org/doi/10.1103/PhysRevLett.104.093602>
- [7] M. Koschorreck, M. Napolitano, B. Dubost, and M. W. Mitchell, "Quantum Nondemolition Measurement of Large-Spin Ensembles by Dynamical Decoupling," *Phys. Rev. Lett.* **105**, 093602 (2010). <http://link.aps.org/doi/10.1103/PhysRevLett.105.093602>

Reply to comments of Reviewer 3

We thank Reviewer 3 for a careful and detailed reading of the manuscript, and for identifying several aspects that required improvement. Below we give a point-by-point response to the Reviewer's comments. Resulting changes are indicated **in blue** in the revised manuscript.

Comment 1. The authors give a bound for the number of entangled particles. I am wondering whether it would be possible to also infer something about the type of entanglement, i.e. two-particle singlets versus more complex multi-party entanglement. I am in particular thinking about the methods used in this paper:

<https://journals.aps.org/prl/abstract/10.1103/PhysRevLett.86.4431>

Could they be adapted to the present experiment? Or is it clear that no multi-party entanglement will be created?

Response: We thank the Reviewer for this question. In the paper indicated, "Entanglement and extreme spin squeezing" by Sørensen and Mølmer, it is shown how to infer multi-partite entanglement from the mean and variances of the total spin of an ensemble. The methods used there only detect multi-partite entanglement in polarized states; when the mean polarization is zero (or small) no entanglement at all is detected. So their methods do not apply to our scenario. It is also easy to show that bipartite entanglement is sufficient to explain a small total variance, i.e. small $\text{var}(F_x) + \text{var}(F_y) + \text{var}(F_z)$: because bipartite singlets have zero variance, a state containing many bipartite singlets can have arbitrarily small total variance, even if it has no multi-partite entanglement. So no method using the ingredients (mean and variance) considered by Sørensen and Mølmer could detect multi-partite entanglement in our system. Some work has been done on detecting multi-partite entanglement with higher-order statistics beyond the mean and variance, e.g.¹⁻³. So there is hope that multi-partite entanglement could some day be detected in this system, but it would require at least a considerably different statistical analysis and possible new kinds of measurements. There is, we think, good reason to believe there is multi-partite entanglement in the singlet: The method of entanglement, passing a probe beam through *all* the atoms, does not in any way enforce which atom should be entangled with which. That is, the coupling to the light field is invariant under permutation of that atoms. The most natural outcome would be for all atoms to be entangled in a very large, permutationally-invariant entangled state.

Comment 2. The authors also estimate the spatial range of entanglement based on applying a magnetic field gradient. They find a range that is much greater than a wavelength, but smaller than the size of the cell. Can this result be understood quantitatively?

Response: The entanglement range has been reported in the paper is the average distance of entanglement bonds. For this reason, it must be smaller than the size of the cell. The result is close to what you would get if you assumed a permutationally invariant entanglement: any atom is equally likely to be entangled with any other. This is what you would expect from the optical method of entanglement (already mentioned in response to the previous question). In this simple model there should be all possible atomic separations, from zero to the length of the cell, with a more likelihood for the shorter separations. The average separation would be a fraction of the length of the cell, which agrees reasonably well with what is observed.

Comment 3. I gained some understanding of the SERF regime from reading this paper, but I do feel that the explanation could still have been more comprehensive and self-contained. Some related detail questions include the size of A_{hf} (it would have been nice to see that somewhere on page 2 or 3), and the meaning of the 'nuclear slowing-down factor' q .

Response: We thank the Reviewer for the very important comment. For ^{87}Rb D_1 line, the hyperfine splitting is around 6.8 GHz, which we have added this information on page 2. We also added one sentence and one reference for nuclear slowing-down factor on page 3.

Comment 4. Finally a minor point, there is a lot of repetition towards the end of the abstract.

Response: We thank the Reviewer for pointing out the repetition in the abstract, which was a typesetting error. We have corrected it in the revised manuscript.

1. J. K. Korbicz, J. I. Cirac, and M. Lewenstein, "Spin squeezing inequalities and entanglement of n qubit states," Phys. Rev. Lett. **95**, 120502 (2005).
2. A. R. U. Devi, R. Prabhu, and A. K. Rajagopal, "Characterizing multiparticle entanglement in symmetric n -qubit states via negativity of covariance matrices," Phys. Rev. Lett. **98**, 060501 (2007).
3. M. Hillery, H. T. Dung, and H. Zheng, "Conditions for entanglement in multipartite systems," Phys. Rev. A **81**, 062322 (2010)

REVIEWERS' COMMENTS:

Reviewer #1 (Remarks to the Author):

Authors' answers clarified questions I have and changes to the manuscript are appropriate. Since the topic is new and active I believe the publishing of the manuscript should be expedited to stimulate the further discussion.

Denis Sukachev

Reviewer #2 (Remarks to the Author):

Carefully reading the manuscript again, and the detailed reply of the authors to the comments and questions of all referees I am convinced that the authors understand the system they have worked with at an extreme level of detail, and what they have achieved is

- 1) an in-depth experimental analysis of a quantum measurement of collective spin in a hot unpolarized atomic vapor in the SERF regime.
- 2) a novel theoretical analysis of the relevant measurement uncertainties.
- 3) a convincing demonstration of the creation of macroscopic entangled states in a system readily amenable to experimentation by a large number of groups.

In particular, the previous achievements clearly connect, to my knowledge for the first time, several communities working on very hot topics, the community of atomic magnetometers with atomic vapors, the community of quantum metrology, the community of theoretical quantum information (in particular those working on quantifying and detecting multi-body high-dimensional entanglement).

I expect this work to trigger a broad "offense" of several workers in the above communities on a very promising system rich in fundamental physics, with the potential to see in the near future several further advances in our understanding of the complicated underlying physics as well as the development of novel quantum sensing technology.

In summary, this work can be clearly characterized as "ground-breaking", hence it should be disseminated in the most broad publication forum, thus I enthusiastically recommend publication at Nature Communications.

I. Kominis
Department of Physics
University of Crete

Reviewer #3 (Remarks to the Author):

I think that the authors have adequately responded to my questions. I support publication.